# CMOS Radio Frequency Energy Harvester (RFEH) with Fully On-Chip Tunable Voltage-Booster for Wideband Sensitivity Enhancement

**DOI:** 10.3390/mi14020392

**Published:** 2023-02-04

**Authors:** Yizhi Li, Jagadheswaran Rajendran, Selvakumar Mariappan, Arvind Singh Rawat, Sofiyah Sal Hamid, Narendra Kumar, Masuri Othman, Arokia Nathan

**Affiliations:** 1Collaborative Microelectronics Design Excellence Centre (CEDEC), Universiti Sains Malaysia, Bayan Lepas 11900, Malaysia; 2Department of Electrical Engineering, Faculty of Engineering, University of Malaya, Kuala Lumpur 50603, Malaysia; 3Institute of Microengineering and Nanoelectronics, National University of Malaysia, Bangi 43600, Malaysia; 4Darwin College, Cambridge University, Cambridge CB3 9EU, UK

**Keywords:** radio frequency energy harvester, CMOS, voltage boosting, wideband, CCDD, 5GNR

## Abstract

Radio frequency energy harvesting (RFEH) is one form of renewable energy harvesting currently seeing widespread popularity because many wireless electronic devices can coordinate their communications via RFEH, especially in CMOS technology. For RFEH, the sensitivity of detecting low-power ambient RF signals is the utmost priority. The voltage boosting mechanisms at the input of the RFEH are typically applied to enhance its sensitivity. However, the bandwidth in which its sensitivity is maintained is very poor. This work implements a tunable voltage boosting (TVB) mechanism fully on-chip in a 3-stage cross-coupled differential drive rectifier (CCDD). The TVB is designed with an interleaved transformer architecture where the primary winding is implemented to the rectifier, while the secondary winding is connected to a MOSFET switch that tunes the inductance of the network. The TVB enables the sensitivity of the rectifier to be maintained at 1V DC output voltage with a minimum deviation of −2 dBm across a wide bandwidth of 3 to 6 GHz of 5G New Radio frequency (5GNR) bands. A DC output voltage of 1 V and a peak PCE of 83% at 3 GHz for −23 dBm input power are achieved. A PCE of more than 50% can be maintained at the sensitivity point of 1 V with the aid of TVB. The proposed CCDD-TVB mechanism enables the CMOS RFEH to be operated for wideband applications with optimum sensitivity, DC output voltage, and efficiency.

## 1. Introduction

Since the 3rd Generation Partnership Project (3GPP) standardized the first New Radio (NR) version (Release 15) in mid-2018, the evolution of the fifth-generation (5G) new radio (NR) has been rapid. 5GNR technology was designed to operate in two distinct bands: FR1 (410 to 7125 MHz) and FR2 (24,250 to 52,600 MHz). Despite operating into the 7 GHz band, FR1 is commonly referred to as the “Sub-6 GHz” band. The leading carriers compete to provide various commercial services over 5G networks. In the future, it is expected that over 6.5 million 5G base stations will be installed, allowing over 58% of the world’s population to access services via over 100 billion 5G connections [1]. Following 5G’s rapid development, many commercialization use cases will push the 5G network to improve performance and expand capabilities continuously.

It is no secret that the worldwide demand for electronic devices has been rising, which means that the global use of electrical energy has been growing alongside it. According to studies cited in [2], by 2030, communication technologies might account for as much as 51% of the world’s total electricity consumption. The study also indicates that by 2030, the power used by communication technologies might account for up to 23% of worldwide greenhouse gas emissions. As a result, reducing the energy needed to run all these gadgets connected to the internet is an absolute necessity. Furthermore, these connected devices are powered by batteries, contributing to increased waste and environmental pollution. Since the introduction of the first commercial lithium-ion (Li-Ion) batteries in 1991, the number of portable electronic device products that use Li-Ion batteries has grown dramatically [3].

Since the battery life of connected electronic devices can be vastly improved by reducing the frequency with which they need to be charged, it is crucial to lower their power consumption. Several energy harvesting works have been successfully implemented recently, including the triboelectric nano-generators (TENG) devices [4,5]. TENG devices are commonly utilized for biomechanical energy harvesting applications. In [4], a nanocomposite that acts as a positive triboelectric layer was successfully used in the fabrication of the TENG device, which converts waste mechanical energy into valuable electrical energy. The TENG device consumes an area of 4cm^2^ while capable of delivering an output voltage of 35 V as well as 130 nA output current at 100 MΩ load. Moreover, in [5], a ZIF-8 HG-Kapton TENG device with dual-mode operation was proposed. A triple-unit TENG is constructed using additive manufacturing, producing an output voltage of 150 V and a current of 4.95 µA. The single-unit mode is implemented to detect a robotic system’s right and left tilting motion. Meanwhile, the triple-unit mode is utilized as a sustainable power source to power up low-power electronics. However, the TENG devices mainly focus on harvesting the wasted mechanical motions, which is not present in on-chip microelectronics integrated circuits.

Radio frequency energy harvesting (RFEH) is a particularly prevalent type of renewable energy harvesting at the moment [6,7,8,9,10,11] because RFEH can be unified into a single wireless communication system used by all interconnected electronic devices. RFEH is widely studied and implemented in integrated circuits to improve the system’s efficiency and charge the batteries. It aids in decreasing the need for large battery packs in electronic gadgets and the amount of time between charges. The RFEH system consists of an antenna, an input impedance matching network, a rectifier, a voltage multiplier, and an energy storage component or load.

The antenna in the RFEH system detects or receives the transmitted RF wave propagating through the transmission medium. Its impedance-matching network is designed to provide maximum power transfer for the RFEH system. The impedance matching network employs low-loss or high Q-factor inductors or transformers. The rectifier converts the received alternating current (AC) RF signal to direct current (DC) power. However, the rectifier’s DC output voltage is too low to power a wireless device. As a result, a voltage multiplier is used to increase the DC voltage level to the level required by wireless devices. Finally, the harvested energy is stored in energy storage devices such as rechargeable batteries and super-capacitors or directly applied to the load. Figure 1 depicts the block diagram of an RFEH system [12].

A reconfigurable RFEH with dual-path rectifiers and an adaptive control circuit (ACC) is presented in [13]. The dual-path rectifiers are configured with series and parallel paths for low-power and high-power operations. This method maintains a high-power PCE over a wide range of input power. The rectifiers use internal threshold voltage cancellation (IVC) to compensate the transistors’ V_th_ effectively. The ACC includes a comparator, an inverter, and three switches that activate series or parallel paths based on the RF input power range. An off-chip impedance matching network is used to match the rectifier’s input impedance to the source of the RF input power. For a 200 kΩ load at 0.902 GHz, the reconfigurable RFEH achieved a peak power conversion efficiency (PCE) of 33% and a DC output voltage of 3.23 V with −8 dBm RF input power. It has a sensitivity of −20.2 dBm input power and outputs 1 V DC voltage for a 1 MΩ load.

Furthermore, a 3-stage Cross-Coupled Differential-Drive (CCDD) rectifier with a broad PCE dynamic range was proposed in [14]. The proposed RFEH uses a self-body biasing approach to lower the V_th_ and diode-tied transistors to lower the reverse leakage current. Both shared-capacitor-coupling (SCC) and individual-capacitor-coupling (ICC) input-capacitor topologies are designed independently for the CCDD rectifier. Two separate capacitors are used to separate the DC power supply from the RF signal route. When used in conjunction with one another, these capacitors bias the gate of the transistors while one store charge. The SCC configuration is achieved when the NMOS and PMOS transistor gates are linked to the same coupling capacitor at the input. In contrast, the gate biasing of each NMOS and PMOS transistor is kept independent by physically separating the coupling capacitor in the ICC arrangement. Input power of −18.4 dBm at 0.9 GHz resulted in a peak PCE of 83.7% for the SCC configuration. At the same frequency, the ICC configuration’s peak PCE was 80.3% when fed with an input power of −17 dBm.

Moreover, an RFEH based on an improved Dickson charge pump with an output capacitor to reduce the load capacitance during the positive half cycle is proposed in [15]. The rectifiers addressed here are modified output capacitor Dickson and differential load Dickson. The Dickson charge pump scheme is used in both rectifiers, which consists of diode-connected transistors, charge storage coupling capacitors, load capacitors for output ripple reduction, and resistive load. The first design incorporates a 3-stage voltage multiplier with a modified output capacitor loop. In the meantime, the second design employs a 2-stage voltage multiplier with a differential load. The output capacitor helps increase the DC output voltage, allowing for appropriate load resistors. The differential load, on the other hand, helps to improve the rectifier’s sensitivity to operate for low RF input signals at 0.953 GHz. Under −12.5 dBm input RF power, the rectifier with a modified output capacitor achieved a peak PCE of 84.4% and 1 V DC output voltage. Meanwhile, under −15 dBm input power, the rectifier with differential capacitor load achieved a peak PCE of 56.2%.

As observed from recent on-chip RFEH works, it is evident that most of them are operating in narrowband applications, and a wideband energy harvesting solution is still a challenge, unlike in other RF circuits, such as power amplifiers or voltage-controlled oscillators, which achieve wideband [16,17]. Thus, in this study, a 3-stage CCDD rectifier is equipped with an on-chip tunable voltage boosting mechanism that is entirely adjustable for wideband sensitivity tuning. The tunable voltage booster is constructed with an interleaved transformer architecture, with the primary winding connected to the rectifier while the secondary winding is connected to a MOSFET switch that tunes the network’s inductance. Adjusting the voltage booster enables the CMOS RFEH’s sensitivity to be maintained at 1V DC output voltage throughout a bandwidth of 3 to 6 GHz of 5GNR frequency bands. The suggested technique allows the CMOS RFEH to operate with optimal sensitivity, DC output voltage, and PCE for wideband applications.

## 2. Rectifier Design and Operation

The rectifier designed adopts the cross-coupled differential-drive (CCDD) architecture, as depicted in Figure 2. Referring to Figure 2, the CCDD rectifier comprises two NMOS transistors (M_N1_ and M_N2_) and two PMOS transistors (M_P1_ and M_P2_) connected in a cross-coupled configuration. The transistors in the rectifier are operating in the subthreshold region, and the drain-current I_D_ is given as [18]:(1)$$I_{D}=I_{S}\cdot e^{\frac{V_{GS}-V_{TH}}{nV_{T}}}\left(1-e^{-\frac{V_{DS}}{V_{T}}}\right)$$
where I_S_ is the zero-bias current of the device, V_GS_ is the gate-source voltage of the transistor, V_DS_ is the drain-source voltage of the transistor, V_TH_ is the threshold voltage of the transistor, n is the subthreshold slope, and V_T_ is the thermal voltage.

M_N1_ and M_N2_ adopt a self-body-biasing mechanism in which their bulk terminals are connected to the DC output voltage (V_DC_). The self-body-biasing mechanism reduces the threshold gate voltage (V_TH_) of M_N1_ and M_N2_. Figure 3 compares drain current (I_D_) with and without bulk biasing. It can be observed that the V_TH_ of the transistor can be mitigated to 440 mV from 540 mV after applying the bias to the bulk. The transistor’s V_TH_ contributes to voltage drop, limiting the maximum V_DC_ and the PCE that can be achieved. Therefore, it is essential to lower the transistor’s V_TH_ for the rectifier. Furthermore, to mitigate the reverse-leakage current, diode-connected MOS transistors (D_1_–D_4_) are integrated between the gate and the source of each transistor, as depicted in Figure 2. The diodes also aid in raising the forward conduction of the rectifier.

Assuming a steady-state operation, when the RF input power of the rectifier is low and for the first half-period of the input RF signal (V_RF+_ > V_RF-_), M_P1_ and M_N2_ are turned on to perform rectification while M_P2_ and M_N1_ are turned off. For the second half-period of the input RF signal, M_P2_ and M_N1_ perform the rectification function while M_P1_ and M_N2_ are turned off. During the first and second half-periods, the bulk of M_N2_ and M_N1_ are, respectively, biased by the V_DC_ from the rectifier in which it realizes the self-body-biasing mechanism. The self-body-biasing is not implemented on M_P1_ and M_P2_ since only the positive DC voltage is produced in the rectifier circuitry. At the low RF input signal, the reverse leakage current is minimal as the resistive loss mainly contributes to the energy loss during forward conduction. This is due to the low DC biasing voltage applied at the transistor’s bulk, which is not significant for V_TH_ reduction. A voltage of more than 500 mV is necessary to overcome the forward voltage of the diode-configured transistors utilized in the rectifier circuit. This is shown in Figure 3, where the I-V curve of the diode configured MOS with W/L of 12.0 µm/0.13 µm begins to conduct current beyond 540 mV. Thus, D_1_–D_4_ is turned off when the RF input signal is low.

On the other hand, when the RF input signal is large enough to overcome the forward voltage of D_1_–D_4_, it starts to operate. At M_N1_ and M_N2_, D_1_ and D_2_ are in reverse biased condition when n-stage V_DC_ > V_RF±_. n-stage V_DC_ is the DC output voltage from the previous stage of the rectifier. The potential difference between the source and gate creates an extra current path for the forward current to flow to the system’s drain, increasing the PCE. Suppose the voltage V_DC_ > V_RF±_, the D_1,_ and D_2_ are in forward biased condition. The diodes act as a large resistor that blocks the reverse-leakage current flowing from n-stage V_DC_ into V_RF±_. Moreover, when V_DC_ < V_RF±_, D_3,_ and D_4,_ which are connected at the M_P2_ and M_P1,_ respectively, are in reverse bias. D_3_ and D_4_ emulate a huge resistor with a potential of –V_D_ between the gate and source of the transistors to block the reverse leakage current from flowing back to the source. Suppose the voltage V_DC_ > V_RF±_, D_3,_ and D_4_ are in forward bias, where it increases the forward current by pushing further charge to the output, increasing the total output charge and thus enhancing V_DC_, sensitivity, and PCE of the rectifier.

As depicted in Figure 2, the proposed CCDD rectifier is implemented with a mutual capacitor coupling configuration as the gates of the NMOS and PMOS are connected to the same coupling capacitor at the input. The capacitors pair (C_2_ and C_3_) decouple from the gates of the transistors to isolate the RF signal path from the power line with the addition of C_1_ and C_4_. C_2_ and C_3_ serve as charge-storing capacitors only, while C_1_ and C_4_ focus solely on biasing the gate of the transistors without being impacted by the DC offset contributed by C_2_ and C_3_. Figure 4 depicts the input RF voltage waveforms for the proposed CCDD rectifier.

When implemented in ambient RFEH systems, cascading more stages typically allows for a higher V_DC_ output. Albeit, as a result of its proportionality to RF input power (P_RFIN_) and PCE for a fixed output load (R_L_), the power equation in (2) imposes a theoretical limit on the possible value of V_DC_. The PCE of the rectifier is defined as in (3). As demonstrated in Figure 5, extending the rectifier stages beyond necessary would be superfluous because V_DC_ would not increase considerably. By referring to Figure 5, it can be deduced that the V_DC_ is not rising significantly beyond 3 stages and begins to drop as the stages increase. Therefore, a 3-stage rectifier is considered essential, and its configuration is depicted in Figure 6. Meanwhile, Figure 7 shows the rectification response of the 3-stage CCDD rectifier at 3 GHz.
(2)$$V_{DC}=\sqrt{\eta \cdot P_{RFIN}\cdot R_{L}}$$
(3)$$\eta =\frac{P_{DC}}{P_{RFIN}}=\frac{V_{DC}^{2}}{R_{L}\cdot P_{RFIN}}\times 100\%$$

## 3. Tunable Voltage Booster (TVB) Design and Operation

The tunable voltage booster (TVB) consists of a primary conductor and a secondary conductor interleaved between each other, as illustrated in Figure 8. The primary conductor is the main inductor connected to the input of the rectifier. In contrast, the secondary conductor is the auxiliary conductor used to tune the inductance of the primary conductor through magnetic coupling. A transistor switch is connected in parallel to the secondary conductor, which opens and shorts the conductor when the gate voltage is applied. A coupling coefficient, k, magnetically couples the primary conductor and secondary conductor. When the ambient RF input signal is applied to the primary conductor, the changing magnetic field in the primary conductor induces an opposite current flow which translates to an opposing magnetic field on the secondary conductor.

The current flow in the secondary conductor is gradually shorted by varying the gate voltage of the transistor switch. When the transistor switch is entirely off, the secondary coil is open. Thus, no current flow occurs, and the inductance value in the primary conductor remains the default. When the transistor switch is gradually turned on via the applied gate voltage, the secondary conductor becomes shorted in, where the current flow opposes a change of the flux. The opposing magnetic field in the secondary conductor cancels out the magnetic field created in the primary conductor. As a result, the decreasing magnetic field reduces the inductance of the primary conductor, which realizes the variability of the inductance value.

Figure 9 shows the schematic of the TVB configuration applied in the CCDD rectifier. The ambient RF input signal is applied at the primary conductor, L_p_. The secondary conductor, L_s_, is connected to the drain and source of an NMOS transistor, acting as a switch. A DC voltage (V_T_) is applied at the gate of the NMOS transistor, where a resistor, R_1_, is connected in series to provide isolation between the AC signal path and the DC signal path. The secondary conductor is open when the transistor is turned off (V_T_ = 0 V). When the gate voltage is gradually increased (V_T_ > 0 V), the secondary conductor is shorted, which induces an opposite current flow that opposes the magnetic field in the primary conductor, as aforementioned. The current flow paths on each conductor are illustrated in Figure 8. The opposite current flow in the secondary reduces the inductance value in the primary conductor due to their magnetic coupling. The following brief analysis, with the aid of Figure 10, shows the relation of primary inductance when the secondary coil is shorted. Referring to Figure 10:
(4)$$V_{1}=j\omega L_{P}I_{1}+j\omega MI_{2}$$
(5)$$V_{2}=j\omega MI_{1}+j\omega L_{s}I_{2}$$
when L_s_ is shorted,
(6)$$V_{2}=0$$

Thus,
(7)$$I_{2}=-\frac{M}{L_{s}}I_{1}$$

Substituting (7) into (4),
(8)$$V_{1}=j\omega L_{P}I_{1}+j\omega M\left(-\frac{M}{L_{s}}I_{1}\right)$$
(9)$$V_{1}=j\omega L_{P}I_{1}+j\omega M\left(-\frac{M}{L_{s}}I_{1}\right)$$

Since M = k$\sqrt{L_{p}L_{s}}$:(10)$$M^{2}=k^{2}L_{p}L_{s}$$

Substituting (10) into (9),
(11)$$V_{1}=j\omega L_{P}\left(1-k^{2}\right)I_{1}$$

Thus, the inductance of L_P_ when the switch is turned on is:(12)$$L_{P,SW=ON}=L_{P}\left(1-k^{2}\right)$$

It can be deduced from (12) that the inductance of L_P_ is reduced when the switch is turned on at the secondary coil.

The TVB is designed and characterized according to the impedance values needed for the rectifier to achieve wideband sensitivity from 3 to 6 GHz. Figure 11a delineates the simulated inductance value variations of the TVB when VT is tuned from 0 to 2.25 V. Referring to Figure 11a, it can be observed that the inductance (L) can be varied from 5 nH to 1.6 nH at 3 GHz. It is also observed that the inductor’s self-resonance frequency (SRF) is increased to a higher frequency when V_T_ is tuned. On the other hand, Figure 11b shows the respective simulated Q-factor of the inductor. This tuning property of the voltage booster is integrated with the rectifier to achieve the wideband functionality of the CCDD rectifier.

## 4. CCDD-TVB Rectifier Design and Operation

The fully integrated architecture of the CCDD-TVB rectifier is illustrated in Figure 12. The rectifier is constructed in 3 stages to achieve the optimum DC output voltage and PCE. The TVB is employed at both differential inputs of the rectifier to provide the tuning property at the rectifier’s input symmetrically. M_N1_ to M_N6_ and M_P1_ to M_P6_ are the NMOS and PMOS transistors utilized in the 3-stage CCDD rectifier. D_1_ to D_12_ are the diode-connected transistors employed to suppress the reverse-leakage current throughout the stages. C_2_, C_3_, C_6_, C_7_, C_10_, and C_11_ are the charge storing capacitors, while C_1_, C_4_, C_5_, C_8_, C_9_, and C_12_ are the gate biasing capacitors. The tunable voltage boosters are L_1_ (L_1P_ and L_1S_), and L_2_ (L_2P_ and L_2S_) applied at the positive (V_RF+_) and negative (V_RF-_) inputs of the rectifier, respectively. L_1P_ and L_1S_ are the primary and secondary coil of L_1_, respectively, while L_2P_ and L_2S_ are the primary and secondary coil of L_2_, respectively. SW_1_ and SW_2_ are the NMOS switches employed at L_1S_ and L_2S,_ respectively, to tune the inductance of L_1P_ and L_2P_. R_1_ and R_2_ are the isolation resistors for both TVBs. The gate voltages for SW_1_ and SW_2_ are applied via V_T1_ and V_T2,_ respectively, to conduct the tuning mechanism of the TVBs. C_L_ is the capacitor load of the rectifier, and it is tested with a resistor load (R_L_) of 250 kΩ.

Conventionally, a series inductor at the rectifier’s input is utilized as the voltage booster (VB). The conventional implementation of the VB has significantly enhanced the V_DC_ of the rectifier. Figure 13 shows the simulated V_DC_ achieved by the rectifier with conventional VB and without VB across the targeted frequency of 3 to 6 GHz. At 3 GHz, the sensitivity of the CCDD rectifier at 1 V V_DC_ is significantly enhanced by −7 dBm (from −17 to −24 dBm). However, when the frequency changes, the sensitivity drastically degrades, reflecting the drawback of the conventional VB. This is due to the change in input impedance after implementing the conventional VB. Observe in Figure 13 that the sensitivity of the rectifier did not vary much without the VB. This is because the input impedance did not change significantly as compared to after VB implementation. It is clearly illustrated in the Smith chart provided in Figure 14. Referring to Figure 14, it can be deduced that the input impedance’s location on the Smith chart changes drastically with the conventional VB compared to without the VB. Since the conventional VB does not have a tuning characteristic, it is a limitation for wideband implementation in rectifiers.

Furthermore, the optimum impedance points for specific parameters can be determined via the load or source pull analysis [19]. Thus, the input impedance for sensitivity across the wideband frequency for the rectifier is identified using the source-pull evaluation. The input impedance for sensitivity at 1 V of less than −20 dBm across frequency is targeted to be achieved. The TVB is designed to fulfill the input impedance needed across the frequency for the targeted sensitivity at 1 V, thus resolving the limitation of the conventional VB, which is narrowband. Figure 15 shows the Smith chart illustrating the input impedance location after tuning the voltage booster. It can be observed that the input impedance’s locations for wideband sensitivity can be attained by tuning the TVB. The TVB provides different values of inductances when tuned as compared to conventional VB, in which it can shift the input impedance locations for sensitivity optimization across wideband frequencies.

After tuning the TVB, the sensitivity of the rectifier at 1 V can be enhanced across the frequency. Figure 16 depicts the V_DC_ achieved by the rectifier before and after tuning. It can be observed that the sensitivity varies drastically by −13 dBm across frequency before tuning. The sensitivity is enhanced to only −2 dBm variation across frequency after tuning the TVB. It proves the functionality of the tunable voltage in realizing the wideband sensitivity rectifier. At 3 GHz, the best V_T_ setting is by default 0 V. At 4 GHz; the V_T_ needed for the optimum performance is 0.25 V. Meanwhile, at 5 and 6 GHz, the V_T_ settings required are 2.25 V and 2 V, respectively.

## 5. Measurement Results

The proposed CCDD-TVB rectifier is fabricated using the technology of CMOS 130 nm and six metal layers. It consumes an area of 1.43 mm^2^ on-chip, including the bond pads for measurement. Figure 17 depicts the fabricated chip of the CCDD-TVB rectifier. In addition, Figure 18 illustrates the measurement setup utilized for validating and characterizing the CCDD-TVB rectifier. The parametric analyzer supplies and monitors the DC voltages (V_T1_ and V_T2_) to the chip. The RF and DC probes are used to probe the connection pads on-chip. Additionally, the multimeter is used to measure the DC output voltage (V_DC_) of the CCDD-TVB rectifier from the PCB variable load. A 250 kΩ load is used as the optimum load for the system. The signal generator is utilized to supply the RF input power (P_IN_) to the rectifier and the RF balun to split the signal into a differential signal.

Figure 19 depicts the measured V_DC_ of the CCDD-TVB rectifier for different load conditions across the P_IN_ at 3 GHz. It can be observed that as the load is increased from 50 kΩ to 250 kΩ, the sensitivity of the rectifier rises as well. The open-load condition is close to the 250 kΩ load, which is required to achieve the highest sensitivity for the rectifier. Thus, 250 kΩ is selected as the optimum load condition required, and the measurement across frequency is conducted. The CCDD-TVB rectifier can deliver a maximum V_DC_ of 2.5 V for P_IN_ of −7 to −8 dBm across the frequency when measured with R_L_ = 250 kΩ. The measured results do not deviate much from simulated results which reflect the reliability of the design, as illustrated in Figure 20. The measured sensitivity at 1 V obtained is from −21.5 dBm at 6 GHz to −23.5 dBm at 3 GHz, where the variation is −2 dBm across the frequency. Figure 21 delineates the sensitivity of the CCDD-TVB rectifier across the frequency of 3 to 6 GHz. The slight deviation of performance between the simulation and measured results is due to the parasitic effect on-chip and process variations during the fabrication of the rectifier.

Figure 22 shows the respective measured PCE achieved by the CCDD-TVB rectifier across the frequency. A peak PCE of 83% is achieved at 3 GHz for −23 dBm P_IN_. At 4 GHz, the peak PCE achieved is 73% for −22.5 dBm P_IN_. At 5 GHz and 6 GHz, the peak PCE is 67% and 57% for −22 dBm and −21.5 dBm, respectively. Referring to Figure 21, the peak PCE is achieved at the lowest sensitivity points across the frequency. It can be observed that the PCE can be maintained at more than 50% across the frequency at its sensitivity point of 1V DC output voltage by tuning the proposed TVB mechanism. Table 1 summarizes the performances of the CCDD-TVB rectifier and compares it to the recent state-of-art rectifiers. It can be observed from the summary that the proposed CCDD-TVB achieves an operating bandwidth of 3 GHz (3 to 6 GHz) as compared to other recent works. This work also realizes the RFEH application for 5GNR bands.

## 6. Conclusions

A wideband sensitivity rectifier is designed and validated in this work. The proposed TVB can tune the input impedance of the designed CCDD rectifier, which realizes the wideband property. The CCDD-TVB rectifier can maintain its sensitivity of 1 V V_DC_ with a minimum deviation of −2 dBm input power across a frequency bandwidth of 3 GHz to 6 GHz. The PCE at the sensitivity point is also maintained by more than 40% across the frequency. A peak PCE of 83% is achieved at 3 GHz for P_IN_ of −20 dBm. All the measured results are validated with a 250 kΩ resistor output load for the rectifier. The maximum DC output voltage of 2.5 V is achieved across the bandwidth. The CCDD-TVB architecture realizes the wideband operating rectifier, which is suitable for 5GNR bands that operate from 3 GHz to 6 GHz.

## Figures and Tables

**Figure 1 micromachines-14-00392-f001:**
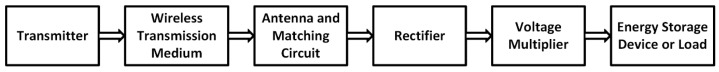
Block diagram of an RFEH system.

**Figure 2 micromachines-14-00392-f002:**
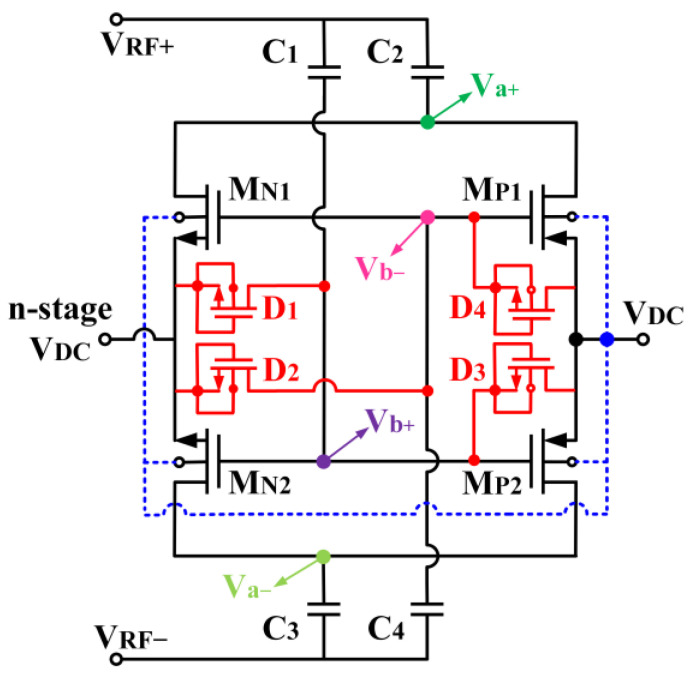
Schematic of the designed CCDD rectifier.

**Figure 3 micromachines-14-00392-f003:**
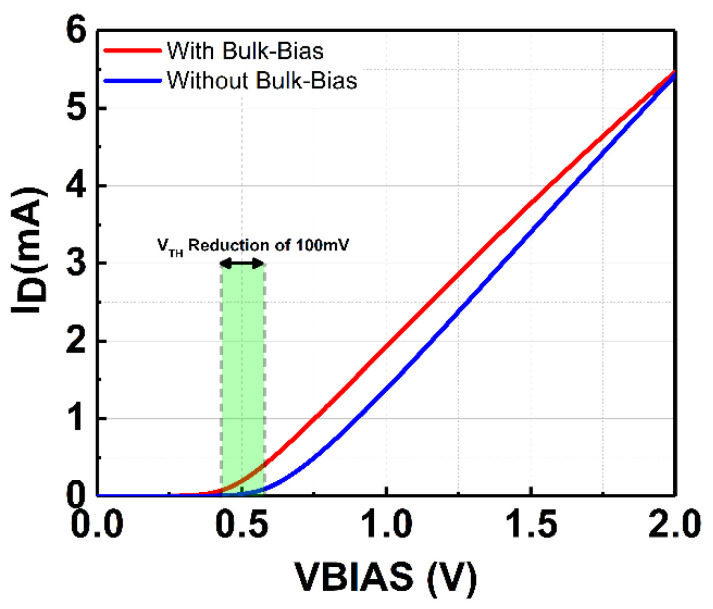
Drain current (I_D_) comparison without and with bulk-biasing. 100 mV V_TH_ reduction is achieved with a bulk-biasing mechanism.

**Figure 4 micromachines-14-00392-f004:**
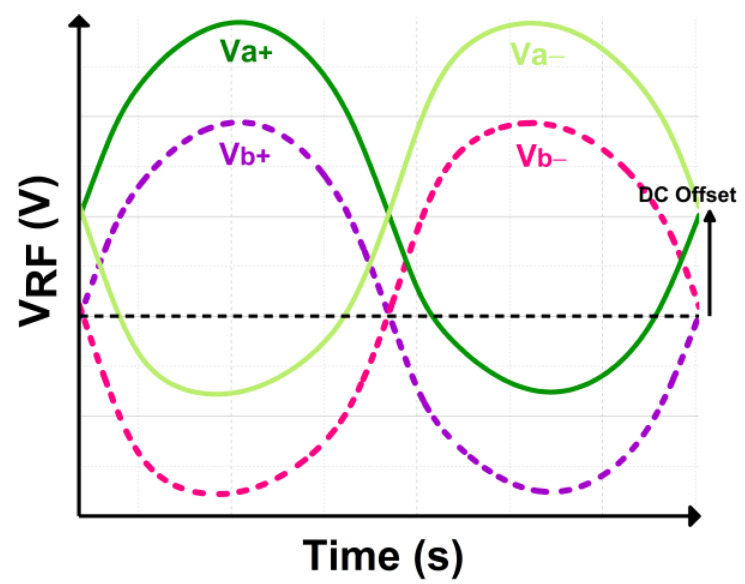
Terminal RF voltage waveform of the designed CCDD rectifier.

**Figure 5 micromachines-14-00392-f005:**
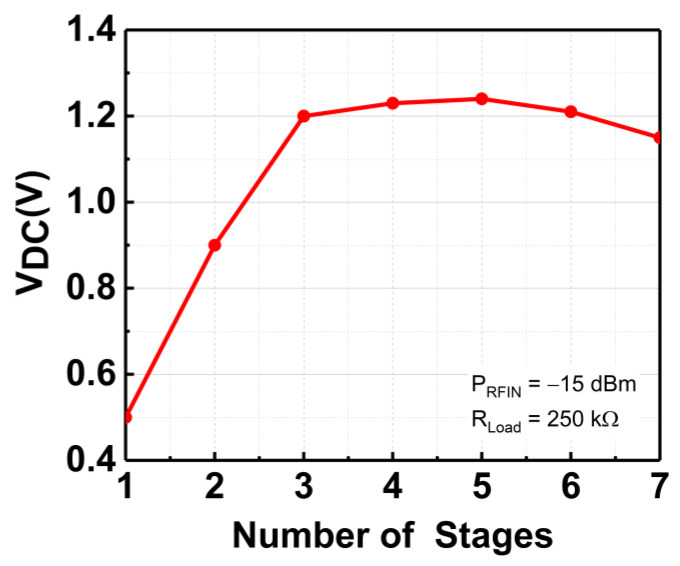
V_DC_ versus rectifier stages.

**Figure 6 micromachines-14-00392-f006:**
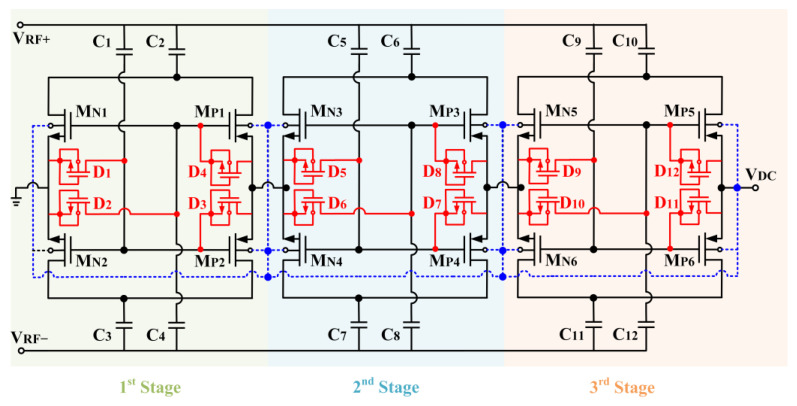
Schematic of the 3-stage CCDD rectifier.

**Figure 7 micromachines-14-00392-f007:**
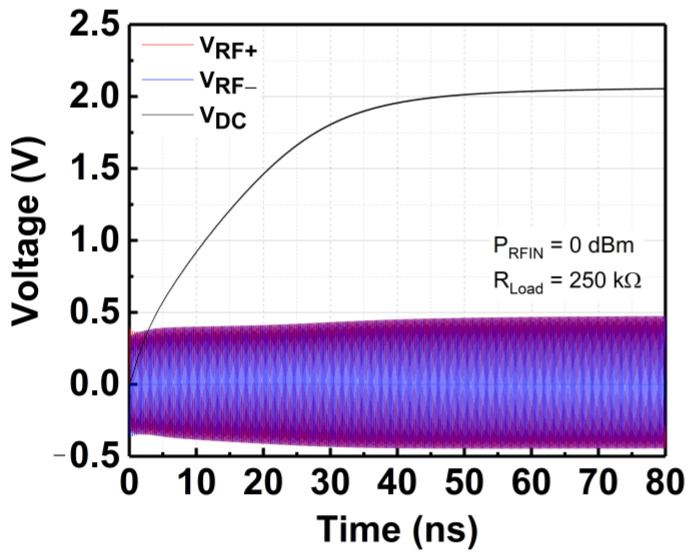
Rectification response of the 3-stage CCDD rectifier at 3 GHz.

**Figure 8 micromachines-14-00392-f008:**
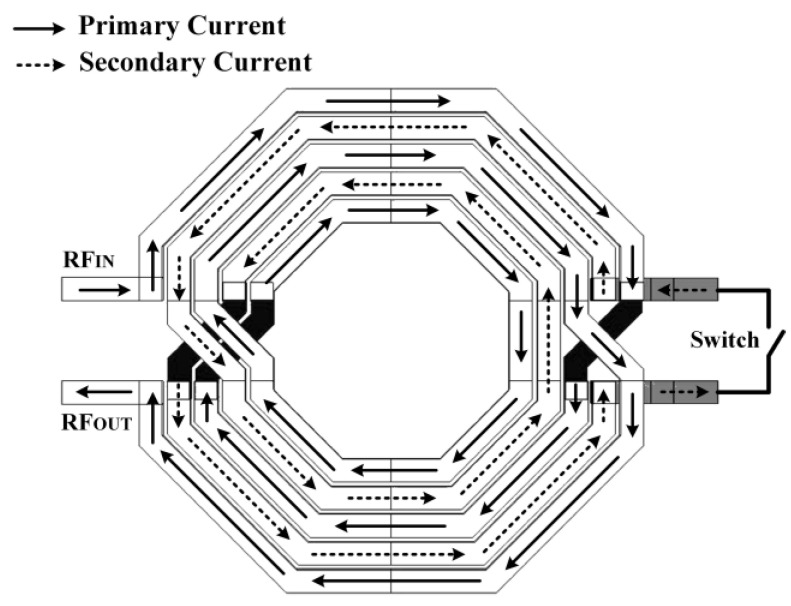
Tunable voltage booster (TVB) architecture.

**Figure 9 micromachines-14-00392-f009:**
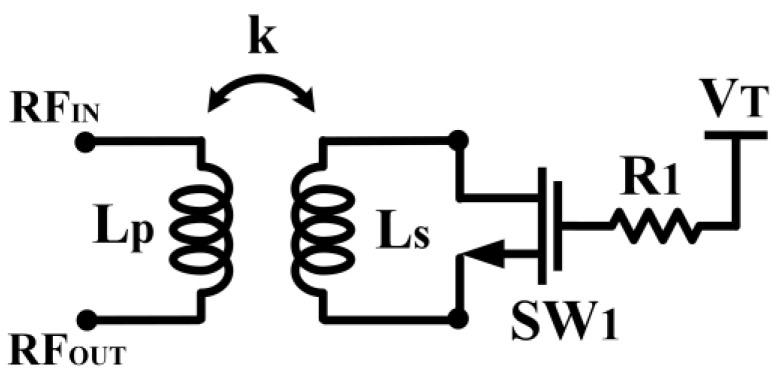
Schematic of the TVB.

**Figure 10 micromachines-14-00392-f010:**
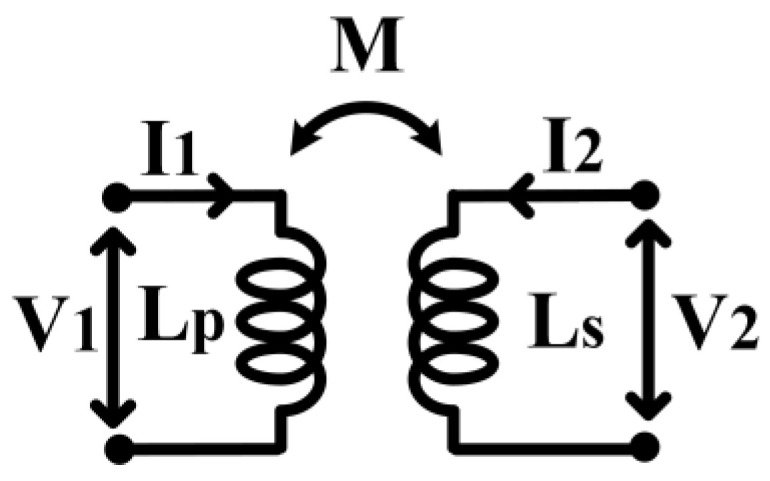
Simple analysis schematic for the TVB operation.

**Figure 11 micromachines-14-00392-f011:**
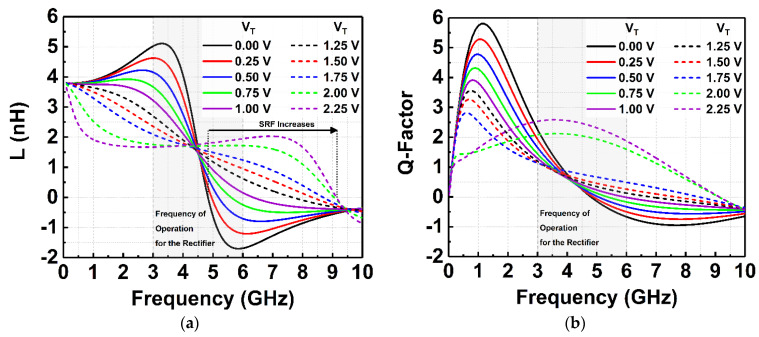
(**a**) Simulated inductance values of the TVB when V_T_ is tuned. (**b**) Simulated Q-factor values of the TVB when V_T_ is tuned.

**Figure 12 micromachines-14-00392-f012:**
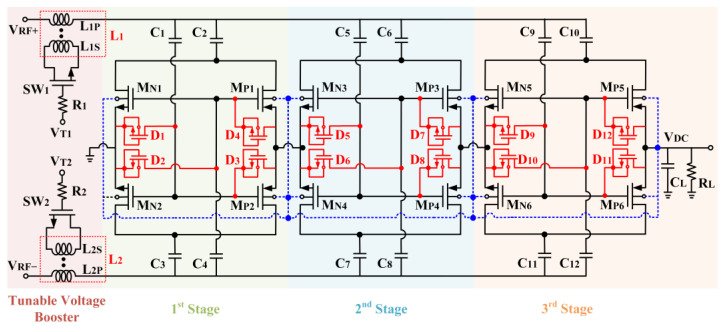
Detailed schematic of the CCDD-TVB rectifier.

**Figure 13 micromachines-14-00392-f013:**
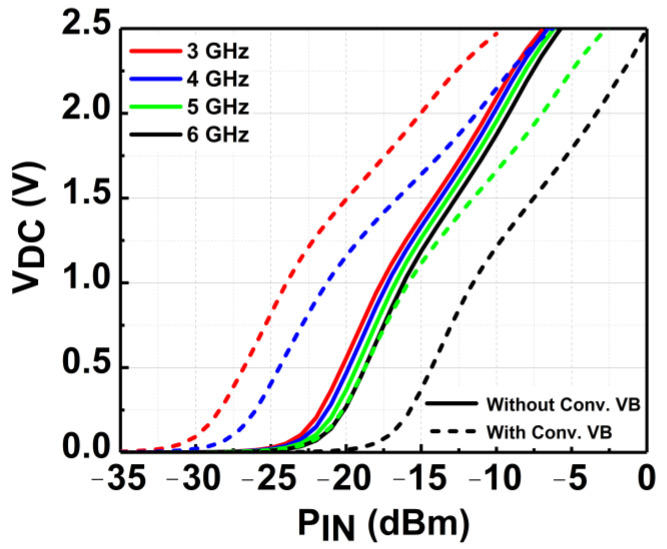
Simulated V_DC_ of the CCDD rectifier with and without the conventional VB.

**Figure 14 micromachines-14-00392-f014:**
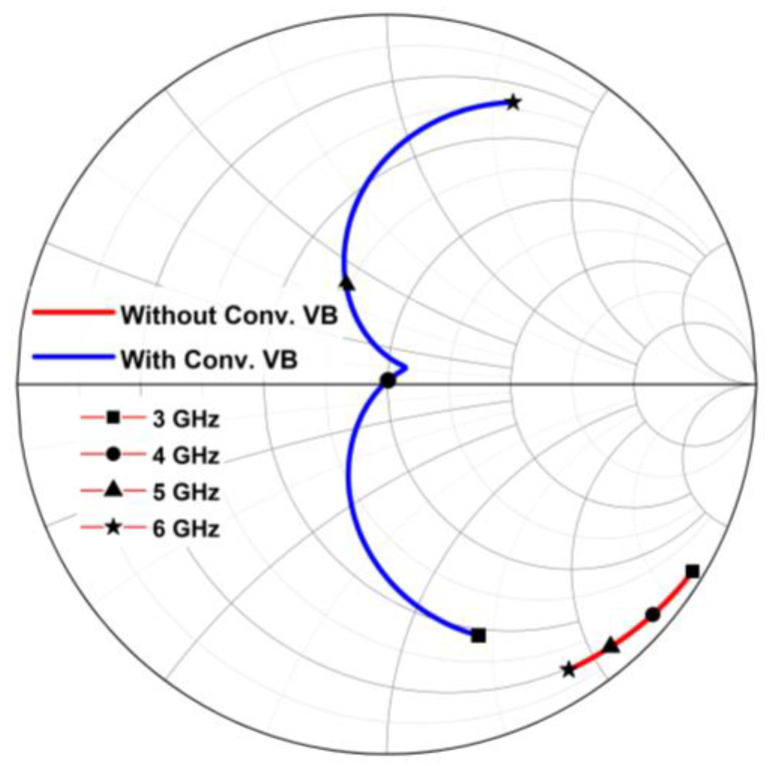
Input impedance’s location across frequency with and without conventional VB.

**Figure 15 micromachines-14-00392-f015:**
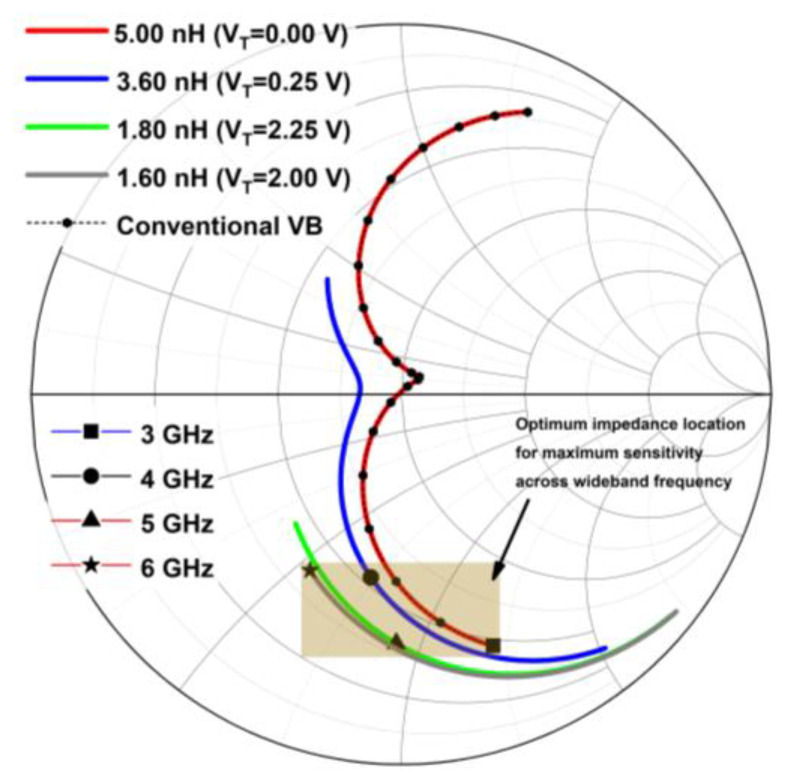
Input impedance’s location across frequency after tuning the TVB as compared to conventional VB.

**Figure 16 micromachines-14-00392-f016:**
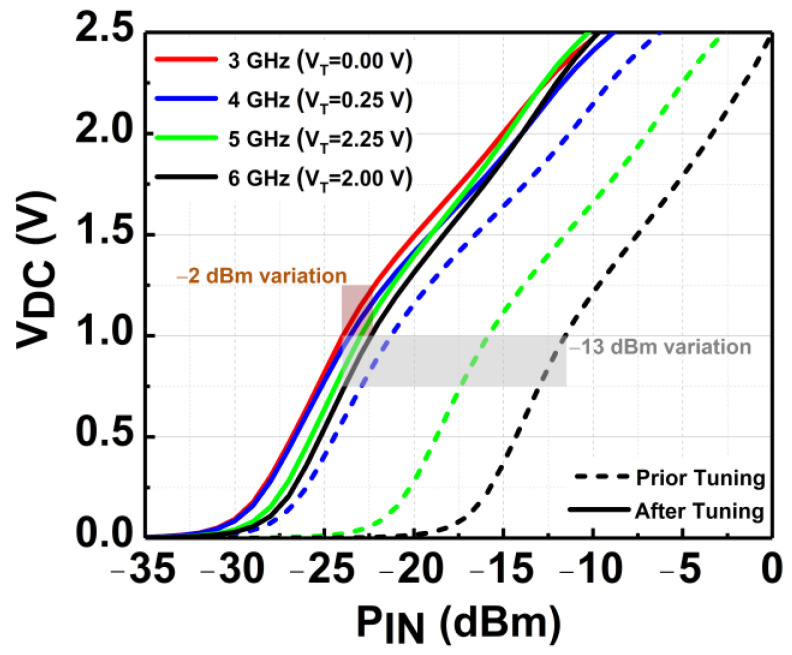
Simulated V_DC_ of the CCDD-TVB rectifier across frequency before and after tuning.

**Figure 17 micromachines-14-00392-f017:**
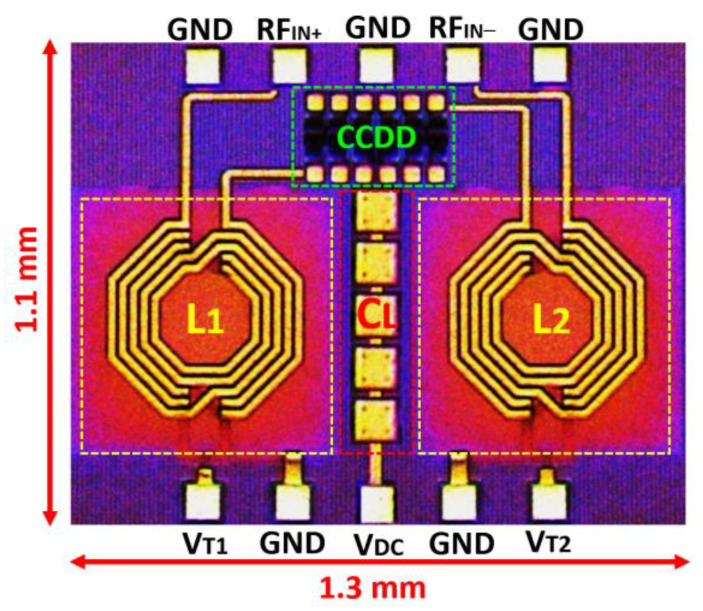
The micro-photograph of the fabricated CMOS 130 nm CCDD-TVB rectifier.

**Figure 18 micromachines-14-00392-f018:**
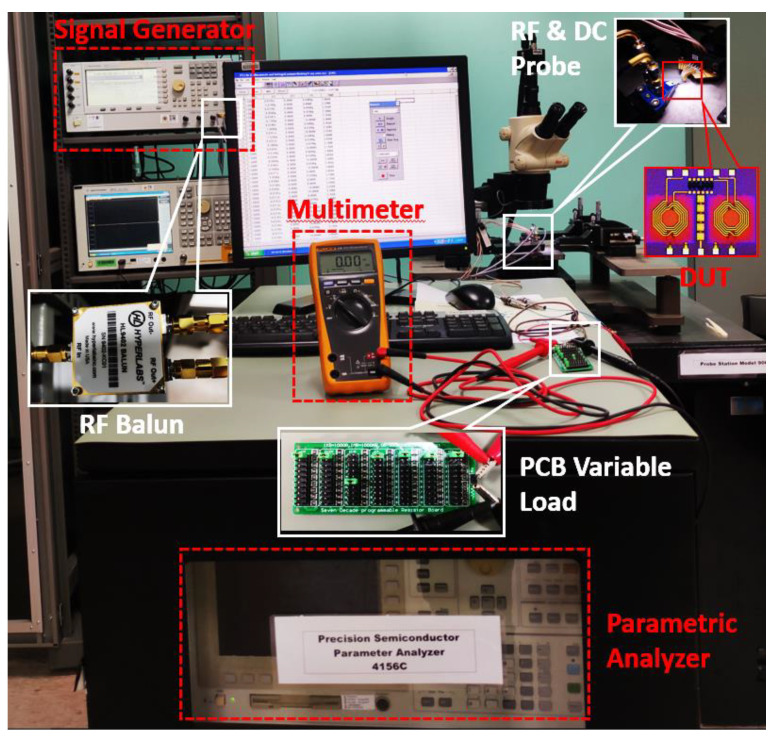
Measurement setup for the validation of CCDD-TVB rectifier.

**Figure 19 micromachines-14-00392-f019:**
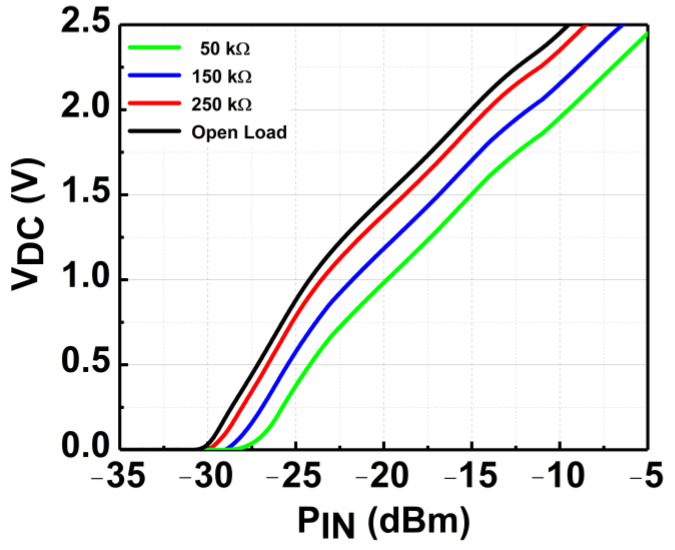
Measured V_DC_ of the CCDD-TVB rectifier at 3 GHz for different load conditions.

**Figure 20 micromachines-14-00392-f020:**
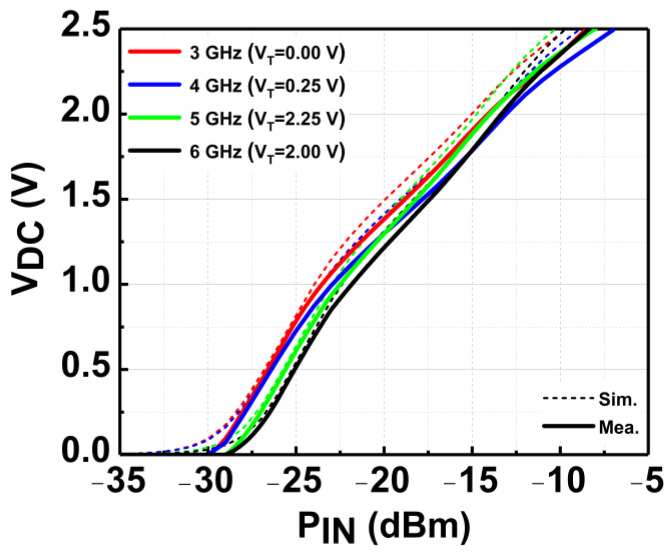
Measured tuned V_DC_ of the CCDD-TVB rectifier across frequency.

**Figure 21 micromachines-14-00392-f021:**
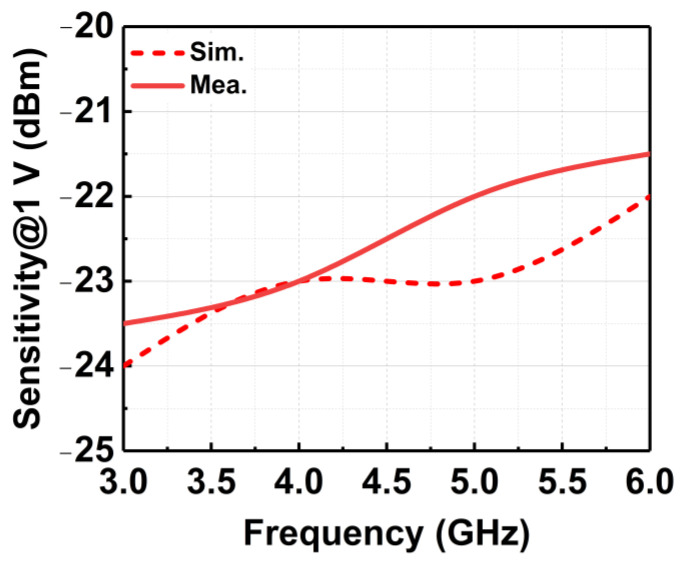
Sensitivity of the CCDD-TVB rectifier across wideband frequency.

**Figure 22 micromachines-14-00392-f022:**
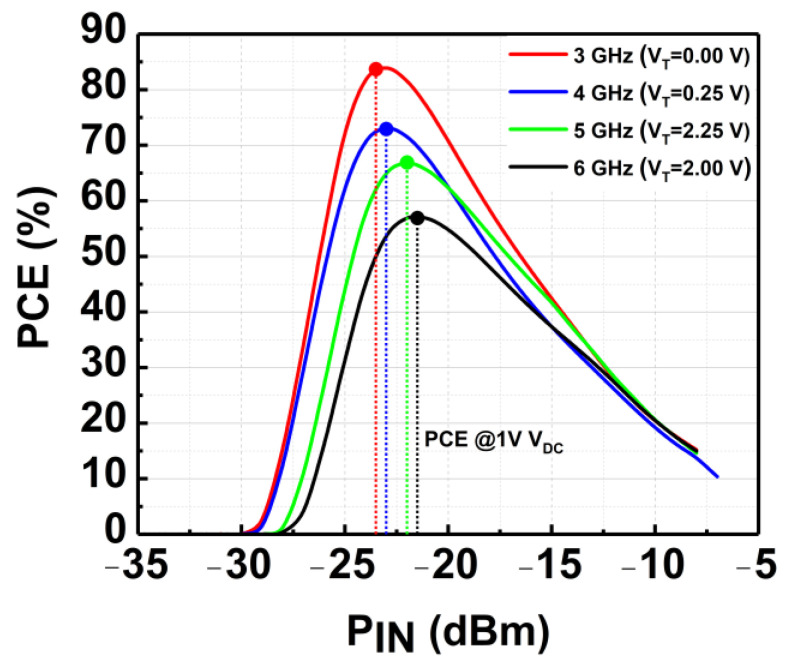
PCE of the CCDD-TVB rectifier across wideband frequency.

**Table 1 micromachines-14-00392-t001:** Comparison of CCDD-TVB rectifier with recent works.

Ref.	Tech.	Rectifier Topology	Freq. (GHz)	Peak PCE (% for Load at Input Level)	VDC_out_ (V)	Sensitivity@VDC (dBm@V)
[13]	180 nm CMOS	1-stage voltage doubler	0.902	33% for 200 kΩ@ −8 dBm	3.23	−20.2@1
[14]	130 nm CMOS	3-stage CCDD	0.9	83.7% for 100 kΩ@ −18.4 dBm	1.1	−19.2@1
[15]	65 nm CMOS	3-stage Dickson	0.953	84.4% for 21.5 kΩ@ −12.5 dBm	1	−12.5@1
[20]	130 nm CMOS	12-stage Dickson	0.915	32% for 1 MΩ@ −15 dBm	3.2	−20.5@1
[21]	180 nm CMOS	Dickson	0.433	30% for 10 kΩ@ −5 dBm	0.5	−9@0.1
[22]	180 nm CMOS	1-stage CCDD	0.1	65% for 100 kΩ@ −18 dBm	1	−18@1
[23]	180 nm CMOS	1-stage CCDD	0.433	65.3% for 50 kΩ@ −15.2 dBm	1	−17@1
[24]	180 nm CMOS	2-stage differential	0.433	74% for 5 kΩ@ −2 dBm	0.8	0@0.8
[25]	180 nm CMOS	5-stage Dickson	0.93/2.63	25.2%/22.5% for 500 kΩ@ −1 dBm	9.5	−16/−15.4@1
[26]	130 nm CMOS	3-stage CCDD Bridge	0.953	72.2% for 10 kΩ@ −1.3 dBm	4.2	−6.3@1
This Work	130 nm CMOS	CCDD + TVB	3–6	83% at 3 GHz for 250 kΩ@ −23 dBm	2.5	−23.5 to −21.5 @1

## Data Availability

Not Applicable.

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
