# Peer review of "CMOS Radio Frequency Energy Harvester (RFEH) with Fully On-Chip Tunable Voltage-Booster for Wideband Sensitivity Enhancement"

_micromachines, 2023, doi:10.3390/mi14020392_

Round 1

Reviewer 1 Report

The manuscript is well written and can be accepted after minor changes. 

Some of my comments are:

1.  Show some Video demonstration of your system "Measurement setup for the validation of CCDD-TVB rectifier" actual working as supporting information?

2. Some important energy harvesting work needs to be added and discussed like Nano Energy 98, 107253, 2022 and Nano Energy 91, 106662, 2021.

3. Some of the data related to measured output dc voltage versus available input power for different load resistance values can be given?

4. English grammar errors need to be corrected carefully. 

Author Response

Dear Professor,

Thank you very much for all the constructive comments. We have addressed all the comments and improve the paper accordingly, except on the video. This is because we performed the measurement in a rented facility and its unavailable at the moment. We humbly apologize regarding this.

Thank you very much.

Best Regards,

Jagadheswaran Rajendran

Reviewer 2 Report

This paper proposes a radio frequency energy harvester with a tunable voltage booster and a 3-stage cross-coupled differential drive rectifier enhancing the wideband sensitivity and overall performance. The experiments were well conducted. Therefore, this work is believed to contribute to the relevant research. It can be considered for publication after addressing the following minor issues:

(1) Few papers are reviewed, the review part in Introduction can be more comprehensive.

(2) The highlights of the paper need to be better extracted to present the innovation and contribution to relevant fields.

(3) Since the tunable voltage booster is the core innovation of proposed harvester, it is better to be compared with other voltage boosters in simulations, not only the situation without booster.

(4) In Fig. 19 and 20, it is suggested to explain the possible factors leading to the errors between simulation and measurement data.

(5) The subject of the paper title is CMOS radio frequency energy harvester, but the research contents mainly focus on the circuit part of harvester. Hence, the other part of the proposed radio frequency energy harvester should be briefly introduced.

Author Response

Dear Professor,

Thank you very much for all the constructive comments. We have addressed all the comments and improve the paper accordingly. Thank you very much.

Best Regards,

Jagadheswaran Rajendran

Reviewer 3 Report

r. In this work, a tunable voltage boost- 19 ing (TVB) mechanism is implemented fully on-chip in a 3-stage cross-coupled differential drive rec- 20 tifier (CCDD). The TVB is designed with an interleaved transformer architecture in where the pri- 21 mary winding is implemented to the rectifier while the secondary winding is connected to a 22 MOSFET switch that tunes the inductance of the network. The TVB enables the sensitivity of the 23 rectifier to be maintained at 1V DC output voltage with minimum deviation of -2 dBm across a wide 24 bandwidth of 3 to 6 GHz of 5G New Radio frequency (5GNR) bands. A DC output voltage of 1 V is 25 achieved as well as peak PCE of 83% at 3 GHz for -23 dBm input power. A PCE of more than 50% 26 is able to be maintained at the sensitivity point of 1 V by the aid of TVB. The proposed CCDD-TVB 27 mechanism enables the CMOS RFEH to be operated for wideband applications with optimum sen- 28 sitivity, DC output voltage and efficiency. Work is well organized and can be considered for publication in current form.

Author Response

Dear Professor,

Thank you very much for the review and comments.

Best Regards,

Jagadheswaran Rajendran

On behalf of all the authors
